# Resistance to ceftazidime–avibactam and other new β-lactams in *Pseudomonas aeruginosa* clinical isolates: a multi-center surveillance study

Felice Valzano,[1] Gianfranco La Bella,[1,2] Teresa Lopizzo,[3] Anna Curci,[3] Laura Lupo,[4] Elisabetta Morelli,[5] Adriana Mosca,[6] Marianna Marangi,[1] Raffaela Melfitano,[7] Tiziana Rollo,[7] Rosella De Nittis,[7] Fabio Arena[1,7,8]

**ABSTRACT**   New β-lactam–β-lactamase inhibitor combinations represent last-resort antibiotics to treat infections caused by multidrug-resistant *Pseudomonas aeruginosa*. Carbapenemase gene acquisition can limit their spectrum of activity, and reports of resistance toward these new molecules are increasing. In this multi-center study, we evaluated the prevalence of resistance to ceftazidime–avibactam (CZA) and comparators among *P. aeruginosa* clinical isolates from bloodstream infections, hospital-acquired or ventilator-associated pneumonia, and urinary tract infections, circulating in Southern Italy. We also investigated the clonality and content of relevant β-lactam resistance mechanisms of CZA-resistant (CZA$^R$) isolates. A total of 120 *P. aeruginosa* isolates were collected. CZA was among the most active β-lactams, retaining susceptibility in the 81.7% of cases, preceded by cefiderocol (95.8%) and followed by ceftolozane–tazobactam (79.2%), meropenem–vaborbactam (76.1%), imipenem–relebactam (75%), and aztreonam (69.6%). Among non-β-lactams, colistin and amikacin were active against 100% and 85.8% of isolates respectively. In CZA$^R$ strains subjected to whole-genome sequencing (*n* = 18), resistance was mainly due to the expression of metallo-β-lactamases (66.6% VIM-type and 5.5% FIM-1), followed by PER-1 (16.6%) and GES-1 (5.5%) extended-spectrum β-lactamases, mostly carried by international high-risk clones (ST111 and ST235). Of note, two strains producing the PER-1 enzyme were resistant to all β-lactams, including cefiderocol. In conclusion, the CZA resistance rate among *P. aeruginosa* clinical isolates in Southern Italy remained low. CZA$^R$ isolates were mostly metallo-β-lactamases producers and belonging to ST111 and ST253 epidemic clones. It is important to implement robust surveillance systems to monitor emergence of new resistance mechanisms and to limit the spread of *P. aeruginosa* high-risk clones.

**IMPORTANCE** Multidrug-resistant *Pseudomonas aeruginosa* infections are a growing threat due to the limited therapeutic options available. Ceftazidime–avibactam (CZA) is among the last-resort antibiotics for the treatment of difficult-to-treat *P. aeruginosa* infections, although resistance due to the acquisition of transferable β-lactamase genes is increasing. With this work, we report that CZA represents a highly active antipseudomonal β-lactam compound (after cefiderocol), and that metallo-β-lactamases (VIM-type) and extended-spectrum β-lactamases (GES and PER-type) production is the major factor underlying CZA resistance in isolates from Southern Italian hospitals. In addition, we reported that such resistance mechanisms were mainly carried by the international high-risk clones ST111 and ST235.

**KEYWORDS**   *Pseudomonas aeruginosa*, ceftazidime-avibactam, β-lactamases, clonal diversity, Southern Italy

Address correspondence to Fabio Arena, fabio.arena@unifg.it.

Felice Valzano and Gianfranco La Bella contributed equally to this article. Author order was determined in order of increasing seniority.

The authors declare no conflict of interest.

See the funding table on p. 10.

*Pseudomonas aeruginosa* is a common cause of infections in both the community and the hospital setting, involving mainly patients with underlying diseases (e.g., diabetic, cystic fibrosis patients and immunocompromised individuals), and contributes to high morbidity and mortality rates (1, 2).

*P. aeruginosa* can display a remarkable array of intrinsic and acquired (mutational or via mobile genetic elements) antibiotic resistance mechanisms (3, 4). Among these, reduced outer membrane permeability due to OprD porin loss, overproduction of the chromosomally encoded AmpC-type β-lactamase or the presence of AmpC variants [*Pseudomonas*-derived cephalosporinase (PDC)], upregulation of the MexAB-OprM/XY efflux systems, and production of acquired β-lactamases [i.e., extended-spectrum β-lactamases (ESBLs) and metallo-β-lactamases (MBLs)] can lead to high level of resistance against a wide range of β-lactams (3–6). ESBLs and MBLs are clinically relevant due to their hydrolyzing activity directed also against the new β-lactam–β-lactamase inhibitor combinations (BL/BLICs), with a consequent drastic reduction of available therapeutic options (5, 7–9). The increasing prevalence of these enzymes is a cause of growing concern, with Imipenemase (IMP)-type, Verona integron-encoded metallo-β-lactamase (VIM)-type, and New Delhi metallo-β-lactamase (NDM)-type MBLs found in all continents (10, 11). For this reason, the discovery and development of new therapeutic strategies that present novel avenues against *P. aeruginosa* infections are increasingly demanded and gaining more attention.

Ceftazidime–avibactam (CZA), a new BL/BLIC combining a third-generation cephalosporin (ceftazidime) and a non-β-lactam β-lactamase inhibitor (avibactam), is active, in most cases, against carbapenem-resistant *P. aeruginosa* when resistance is mediated by porin loss, overexpression of efflux systems, or overproduction of intrinsic AmpC. CZA generally retains activity against strains producing Ambler class A β-lactamases [including Guiana extended-spectrum (GES)-type ESBLs, and *Klebsiella pneumoniae* carbapenemases (KPCs)], class C (i.e., AmpC cephalosporinases), and some class D (i.e., oxacillinase-48) enzymes, whereas it is not active against MBLs (e.g., NDM, VIM, and IMP types) (12–14). Nevertheless, CZA therapy has been associated with good clinical outcomes when used for the treatment of infections caused by *P. aeruginosa* isolates exhibiting a multidrug-resistant (MDR) phenotype (i.e., resistant to at least one tested antibiotic in three or more antimicrobial classes) (15–17).

Another antipseudomonal cephalosporin–β-lactamase inhibitor combination, ceftolozane–tazobactam (C/T), shows a potent *in vitro* activity against *P. aeruginosa*. Ceftolozane has both a high affinity for the penicillin-binding proteins (PBPs) and high stability against various resistance mechanisms (i.e., efflux pumps overexpression, OprD modifications, and AmpC overproduction), whereas tazobactam inhibits class A ESBLs (18, 19). Consequently, C/T represents a valuable option for the treatment of MDR *P. aeruginosa* infections, including those caused by ESBLs producers (20).

Among other novel antibiotics, meropenem–vaborbactam (MVB) and imipenem–relebactam (I-R), two broad spectrum carbapenems combined with two novel β-lactamase inhibitors, are active against Ambler class A and C β-lactamases, with an excellent *in vitro* activity against KPC-producing strains, but ineffective against MBLs or oxacillinases with carbapenemase activity (21–24). However, the activity of MVB against *P. aeruginosa* strains is similar to that of meropenem alone because main carbapenem resistance mechanisms (i.e., porin mutations or upregulation of efflux pumps) are not antagonized by vaborbactam (21, 22). By contrast, relebactam can restore imipenem susceptibility in case of carbapenem-resistant *P. aeruginosa* due to lack of OprD and production of inducible/derepressed AmpC (23, 24).

Because of its high stability against β-lactamases (including carbapenemases, MBLs, and AmpC) and potent activity against MDR *P. aeruginosa* strains, cefiderocol, a novel siderophore cephalosporin, shows considerable potential for the management of difficult-to-threat infections, particularly in the treatment of carbapenem-resistant strains (25–27). Susceptibility testing to cefiderocol is advised because resistant *P. aeruginosa* isolates have been described (28, 29).

In this study, we investigated the activity of CZA and comparator drugs against a recent collection of *P. aeruginosa* strains isolated from five regional hospitals in Southern Italy. We also analyzed the mechanisms that confer resistance to CZA and characterized the CZA-resistant isolates (circulating clones), thus contributing to the epidemiology of the region.

## MATERIAL AND METHODS

### Bacterial isolates

For 6 months (from October 2022 to April 2023), consecutive non-replicate *P. aeruginosa* isolates from cases of bloodstream infections (BSIs), hospital-acquired (HAP) or ventilator-associated pneumonia (VAP), and urinary tract infections (UTIs) were collected and stored at −80°C by five microbiology laboratories distributed across Southern Italy. Isolates from patients with cystic fibrosis or pediatrics (i.e., age <18 years) were excluded. At the end of the collection period, the isolates were sent to the central laboratory for confirmation of identification at the species level by matrix-assisted laser desorption ionization–time of flight mass spectrometry (MALDI-TOF MS) (Bruker Daltonics, Billerica, MA, USA), antimicrobial susceptibility testing, and characterization by next-generation sequencing.

Participating laboratories were distributed as follows: two laboratories from two regional teaching hospitals in Northern (i.e., the central laboratory; C1) and Central Apulia (C2); two laboratories from two district hospitals in Central (C3) and Southern Apulia (C4); and one laboratory from a regional hospital in Basilicata (C5).

### Antimicrobial susceptibility testing

At the central laboratory, stored isolates were grown for 18–20 h on non-selective agar medium. Antimicrobial susceptibility to ceftazidime–avibactam and comparator molecules was determined using reference broth microdilution method (30), except for cefiderocol (30 µg disk content, Liofilchem) susceptibility testing where disk diffusion method was used (31).

MICs (and inhibition zone diameters for cefiderocol) were interpreted according to the most recent EUCAST clinical breakpoints (v 13.1, 2023) (32). The reference strain *P. aeruginosa* ATCC 27853 was used as a quality control strain.

### Resistome and clonal relatedness analysis by whole-genome sequencing

Thirty isolates were subjected to whole-genome sequencing (WGS) for the characterization of relevant β-lactam resistance mechanisms [i.e., acquired carbapenemases, such as VIM, IMP, NDM, Florence imipenemase (FIM)-1, and KPC types; and acquired extended-spectrum β-lactamases, such as *Pseudomonas* extended resistant (PER), Vietnamese extended-spectrum β-lactamase (VEB), and GES types]] and clonal analysis. These strains were selected according to the following criteria: (i) resistance to ceftazidime–avibactam (CZA^R); (ii) susceptibility to CZA, and resistance to ceftolozane–tazobactam and/or at least one carbapenem (including the carbapenem–β-lactamase inhibitor combinations). A subset of randomly selected multi-susceptible isolates was also included for comparison.

Some strains fulfilling criteria i) and ii) [four strains for criteria i) and seven strains for criteria ii)] were not viable after storage and thawing and were not included in sequencing experiments.

Bacterial DNA was extracted using an automated MagCore HF16 Plus System with the MagCore Genomic DNA Bacterial Kit (RBC Bioscience, Taipei, Taiwan), according to the manufacturer's recommendations. Genomic DNA was subjected to WGS by the Illumina (San Diego, CA, USA) NovaSeq 6000 platform, using a 2 × 150 bp paired-end reads approach. The quality of sequence read data was checked using Falco (33). Raw reads were assembled using SPAdes (34), using the "--cov-cutoff" flag set to "auto". Raw coverage of the assembled genomes was calculated using a total genome length of 6.3

Mbp corresponding to that of *P. aeruginosa* reference strain PAO1 (GenBank accession no. NC_002516) (35) and ranged between 104× and 195×, with an average value of 142×. A mean of 185 contigs per bacterial genome was obtained, with an average N50 of 385 kbp.

Draft genomes were used to investigate the antimicrobial resistance gene content using the ABRicate software version 1.0.1 (available at https://github.com/tseemann/ABRICATE) with the ARGannot database (36). Prediction of β-lactamase genes was further improved using the NCBI BLASTn tool in an attempt to identify the β-lactamase variants. Molecular typing was performed by the determination of multilocus sequence type (MLST) profile and O types using the *P. aeruginosa* MLST database (available at https://pubmlst.org/paeruginosa) (37) and the *P. aeruginosa* serotyper (PAst) tool (available at https://cge.food.dtu.dk/services/PAst/) (38), respectively. The phylogenetic relatedness was evaluated with the CSI phylogeny 1.4 (available at https://cge.food.dtu.dk/services/CSIPhylogeny/) (39) using default parameters, except for the minimum distance between SNP option, which was disabled. The SNP matrix and phylogenetic trees of the most relevant clones were constructed using *P. aeruginosa* PAO1 genome (GenBank accession no. NC_002516) as reference and draft assembled genomes as input data, after removing contigs <300 bp. The mean percentage of reference genome covered by the isolates belonging to the same sequence type ranged between 94.5% and 95.3%. Phylogenetic trees were visualized and modified by FigTree 1.4.4 (available at http://tree.bio.ed.ac.uk/software/figtree/).

## Study of the genetic context of the *bla*GES-1 gene

The presence of a strong promoter (PcS) [−35 (TTGACA) and −10 (TAAACT)] in the class one integron (IntI1) located upstream the *bla*GES-1 gene and involved in overexpression of GES-1 enzyme causing CZA resistance (40) was investigated by manual analysis using the Basic Local Alignment Search Tool (BLAST). For this purpose, the complete genome of *P. aeruginosa* PSA9 was obtained by combining results from Illumina with those obtained using the Oxford Nanopore Technologies (Oxford, UK) MinION platform, and *de novo* assembly was generated using Unicycler v0.5.0 (41). The integron-encoded antibiotic resistance gene cassette carrying the *bla*GES-1 gene in *P. aeruginosa* PSA9 (GenBank accession no. CP150132) was compared with those of the previously described CZA$^R$ *P. aeruginosa* strains PA5083 (GenBank accession no. CP102174), SE5352 (GenBank accession no. CP054843), and 1903031130 (GenBank accession no. CP060392), which carry a PcS involved in the overexpression of GES-type β-lactamase contributing to resistance to CZA (40).

## RESULTS AND DISCUSSION

### Antimicrobial activity of ceftazidime–avibactam and comparators against *P. aeruginosa* isolates

Overall, 120 non-replicate *P. aeruginosa* clinical isolates were collected by participating laboratories. The number of strains ranged from 2 to 40 in different laboratories. Overall, 62 isolates (51.7%) were from BSIs, 39 (32.5%) from HAP/VAP, and 19 (15.8%) from UTIs.

Colistin was the most active agent (100% susceptibility), followed by cefiderocol (95.8%), amikacin (85.8% susceptibility), ceftazidime–avibactam (81.7% susceptibility), and ceftolozane-tazobactam (79.2% susceptibility) (Table 1). Seven isolates (i.e., *P. aeruginosa* PSA1, PSA3, PSA7, PSA9, PSA39, PSA41, and PSA120; Table S1) were resistant to all β-lactams tested, including the new β-lactam–β-lactamase inhibitor combinations (i.e., ceftazidime–avibactam, ceftolozane–tazobactam, imipenem–relebactam, and meropenem–vaborbactam), except for cefiderocol that remained active in five of seven cases (Table S1).

CZA retained activity against 15 of the 37 isolates that were non-susceptible to ceftazidime (Table S1). Moreover, CZA was active against 39 isolates non-susceptible to at least one β-lactam tested and against five isolates that were also resistant to amikacin and/or tobramycin (Table S1).

**TABLE 1** Numbers and proportion of *P. aeruginosa* strains included in the study and categorized as resistant, susceptible, increased exposure, and susceptible to ceftazidime–avibactam and comparators, and relative $MIC_{50}$ and $MIC_{90}$ (mg/L)

| Molecules[a] | Total isolates screened | No. of isolates categorized as resistant (%) | | No. of isolates categorized as susceptible, increased exposure (%) | | No. of isolates categorized as susceptible (%) | | $MIC_{50}$ (mg/L) | $MIC_{90}$ (mg/L) |
|---|---|---|---|---|---|---|---|---|---|
| AMK[b] | 120 | 17 | (14.2) | 0 | | 103 | (85.8) | 4 | 32 |
| ATM | 92 | 28 | (30.4) | 64 | (69.6) | 0 | | 8 | >32 |
| FEP | 120 | 38 | (31.7) | 82 | (68.3) | 0 | | 4 | >8 |
| CAZ | 120 | 37 | (30.8) | 83 | (69.2) | 0 | | 4 | >32 |
| CZA | 120 | 22 | (18.3) | 0 | | 98 | (81.7) | 4 | >16 |
| C/T | 120 | 25 | (20.8) | 0 | | 95 | (79.2) | 1 | >8 |
| CST[b] | 120 | 0 | | 0 | | 120 | (100) | 1 | 2 |
| IPM | 111 | 39 | (35.1) | 72 | (64.9) | 0 | | 2 | >8 |
| I-R | 84 | 21 | (25.0) | 0 | | 63 | (75.0) | 0.5 | >8 |
| MEM | 120 | 30 | (25.0) | 11 | (9.2) | 79 | (65.8) | 1 | >16 |
| MVB | 92 | 22 | (23.9) | 0 | | 70 | (76.1) | 0.5 | >16 |
| TZP | 120 | 43 | (35.8) | 77 | (64.2) | 0 | | 8 | >32 |
| TOB | 88 | 22 | (25.0) | 0 | | 66 | (75.0) | ≤0.5 | >4 |
| FDC[c] | 120 | 5 | (4.2) | -[d] | | 115 | (95.8) | - | - |

[a]AMK, amikacin; ATM, aztreonam; FEP, cefepime; CZA, ceftazidime–avibactam (avibactam at fixed concentration of 4 mg/L); C/T, ceftolozane–tazobactam (tazobactam at fixed concentration of 4 mg/L); CST, colistin; IPM, imipenem; I-R, imipenem–relebactam (relebactam at fixed concentration of 4 mg/L); MEM, meropenem; MVB, meropenem–vaborbactam (vaborbactam at fixed concentration of 8 mg/L); TZP, piperacillin–tazobactam (tazobactam at fixed concentration of 4 mg/L); TOB, tobramycin; FDC, cefiderocol.
[b]For isolates recovered from infections other than UTI, interpretation criteria refer to the use of the agent in combination with another active agent or measure.
[c]According to disk diffusion method.
[d]No. of isolates categorized as susceptible, increased exposure, as well as data on "MIC50" and "MIC90".

Some differences were observed in the antimicrobial susceptibility pattern comparing isolates from BSI, HAP/VAP, and UTI, with the former on average less resistant to antibiotics tested (Table S1).

Overall, these results showed that the resistance to CZA remained low (i.e., 22 of 120 isolates; 18.3%) among all *P. aeruginosa* isolates included in the study. In fact, CZA was the second most active compound (after cefiderocol) compared with the other β-lactams and β-lactam–β-lactamase inhibitor combinations currently used to treat *P. aeruginosa* infections, such as ceftazidime, meropenem, imipenem, piperacillin–tazobactam, meropenem–vaborbactam, and imipenem–relebactam (42, 43). These results are in accordance with previous studies, where CZA was the most active compound against *P. aeruginosa* after colistin (44–48), thus confirming its potential role in the management of serious and complicated *P. aeruginosa* infections, including those caused by MDR organisms.

## β-lactamase and clonal characteristics of ceftazidime–avibactam-resistant *P. aeruginosa* isolates

WGS was performed to detect both β-lactamase content and clonal relatedness of 18/22 CZA[R] *P. aeruginosa* isolates. The analysis showed that the majority of CZA[R] strains (72.2%, 13/18) carried MBLs, with *bla*$_{VIM-2}$ being the most frequent variant (76.9%, 10/13), followed by *bla*$_{VIM-1}$ and *bla*$_{FIM-1}$ genes that were detected in two and one case, respectively (Table 2). These results are in accordance with previous studies demonstrating that class B β-lactamases are the main underlying mechanism of CZA resistance in *P. aeruginosa*, with VIM-type (mainly VIM-1 and VIM-2) MBLs being the most prevalent (7, 49, 50). A previous nationwide Italian survey on molecular epidemiology of *P. aeruginosa* causing BSI and HAP/VAP showed a proportion of carbapenemase-producing *P. aeruginosa* of 5.1%, with almost 90% accounting for MBL (51). In our study, the overall proportion of carbapenemase-producing *P. aeruginosa* was higher (i.e., 16/120, 13.3%).

Interestingly, all strains carrying VIM-2 carbapenemase belonged to ST111 and were characterized by an identical β-lactamase content since they carried the *bla*$_{OXA-395}$

**TABLE 2** Features of the 30 *P. aeruginosa* strains subjected to whole-genome sequencing analysis

| Strain | Centre code | Origin[a] | Selection criteria[b] | ST[c] | O type | Transferable β-lactamase-coding genes[d] | PDC variants[e] | CZA MIC (mg/L)[f] |
|---|---|---|---|---|---|---|---|---|
| PSA1 | C2 | HAP/VAP | CZA$^R$ | ST235 | O11 | $bla_{PER-1}$, $bla_{OXA-2}$, $bla_{OXA-488}$ | PDC-195 | **>16** |
| PSA3 | C2 | UTI | CZA$^R$ | ST111 | O12 | $bla_{VIM-2}$, $bla_{OXA-395}$ | PDC-216 | **>16** |
| PSA4 | C2 | UTI | CZA$^R$ | ST111 | O12 | $bla_{VIM-2}$, $bla_{OXA-395}$ | PDC-216 | **>16** |
| PSA7 | C2 | HAP/VAP | CZA$^R$ | ST235 | O11 | $bla_{PER-1}$, $bla_{OXA-2}$, $bla_{OXA-488}$ | PDC-195 | **>16** |
| PSA8 | C2 | HAP/VAP | CZA$^R$ | ST235 | O11 | $bla_{FIM-1}$, $bla_{OXA-205}$, $bla_{OXA-488}$ | PDC-195 | **>16** |
| PSA9 | C2 | UTI | CZA$^R$ | ST235 | O11 | $bla_{GES-1}$, $bla_{OXA-488}$ | PDC-195 | **16** |
| PSA10 | C2 | UTI | CZA$^R$ | ST664 | O2 | $bla_{OXA-14}$, $bla_{OXA-50}$ | PDC-119 | **>16** |
| PSA14 | C2 | BSI | CZA$^R$ | ST111 | O12 | $bla_{VIM-2}$, $bla_{OXA-395}$ | PDC-216 | **16** |
| PSA38 | C5 | HAP/VAP | CZA$^R$ | ST643 | O5 | $bla_{PER-1}$, $bla_{OXA-847}$ | PDC-212 | **16** |
| PSA39 | C5 | HAP/VAP | CZA$^R$ | ST235 | O11 | $bla_{VIM-1}$, $bla_{OXA-488}$ | PDC-195 | **>16** |
| PSA41 | C5 | BSI | CZA$^R$ | ST235 | O11 | $bla_{VIM-1}$, $bla_{OXA-488}$ | PDC-195 | **>16** |
| PSA49 | C4 | BSI | CZA$^R$ | ST111 | O12 | $bla_{VIM-2}$, $bla_{OXA-395}$ | PDC-216 | **>16** |
| PSA51 | C4 | BSI | CZA$^R$ | ST111 | O12 | $bla_{VIM-2}$, $bla_{OXA-395}$ | PDC-216 | **>16** |
| PSA62 | C4 | BSI | CZA$^R$ | ST111 | O12 | $bla_{VIM-2}$, $bla_{OXA-395}$ | PDC-216 | **16** |
| PSA70 | C4 | HAP/VAP | CZA$^R$ | ST111 | O12 | $bla_{VIM-2}$, $bla_{OXA-395}$ | PDC-216 | **>16** |
| PSA71 | C4 | HAP/VAP | CZA$^R$ | ST111 | O12 | $bla_{VIM-2}$, $bla_{OXA-395}$ | PDC-216 | **16** |
| PSA72 | C4 | HAP/VAP | CZA$^R$ | ST111 | O12 | $bla_{VIM-2}$, $bla_{OXA-395}$ | PDC-216 | **16** |
| PSA73 | C4 | UTI | CZA$^R$ | ST111 | O12 | $bla_{VIM-2}$, $bla_{OXA-395}$ | PDC-216 | **>16** |
| PSA5 | C2 | HAP/VAP | Non-CZA β-lactam(s)$^R$ [ATM$^R$, FEP$^R$, IPM$^R$, I-R$^R$, MEM$^R$, TZP$^R$] | ST253 | O10 | $bla_{OXA-488}$ | PDC-195 | 8 |
| PSA23 | C2 | BSI | IPM$^R$ | ST298 | O11 | $bla_{OXA-848}$ | PDC-119 | 2 |
| PSA28 | C2 | BSI | Non-CZA β-lactam(s)$^R$ [ATM$^R$, FEP$^R$, C/T$^R$, TZP$^R$] | ST555 | O6 | $bla_{OXA-486}$ | PDC-119 | 4 |
| PSA37 | C2 | HAP/VAP | Non-CZA β-lactam(s)$^R$ [ATM$^R$, IPM$^R$, I-R$^R$, TZP$^R$] | ST260 | O6 | $bla_{OXA-904}$ | PDC-119 | 4 |
| PSA50 | C4 | BSI | Non-CZA β-lactam(s)$^R$ [ATM$^R$, FEP$^R$, IPM$^R$, MEM$^R$, MVB$^R$, TZP$^R$] | ST175 | O4 | $bla_{OXA-50}$ | PDC-216 | 8 |
| PSA60 | C4 | BSI | Non-CZA β-lactam(s)$^R$ [IPM$^R$] | ST308 | O11 | $bla_{OXA-488}$ | PDC-195 | 4 |
| PSA63 | C4 | HAP/VAP | Non-CZA β-lactam(s)$^R$ [IPM$^R$] | ST308 | O11 | $bla_{OXA-488}$ | PDC-195 | 2 |
| PSA64 | C4 | HAP/VAP | Non-CZA β-lactam(s)$^R$ [IPM$^R$] | ST2104 | O6 | $bla_{OXA-851}$ | PDC-208 | 1 |
| PSA29 | C2 | BSI | Non-resistant pattern | ST1033 | O4 | $bla_{OXA-50}$ | PDC-212 | 2 |
| PSA44 | C5 | HAP/VAP | Non-resistant pattern | ST1248 | O1 | $bla_{OXA-395}$ | PDC-195 | 1 |
| PSA56 | C4 | BSI | Non-resistant pattern | ST381 | O5 | $bla_{OXA-50}$ | PDC-119 | 2 |
| PSA79 | C1 | HAP/VAP | Non-resistant pattern | ST571 | O11 | $bla_{OXA-494}$ | PDC-212 | 0.5 |

[a]BSI, bloodstream infection; HAP/VAP, hospital-acquired/ventilator-associated pneumonia; UTI, urinary tract infection.
[b]CZA$^R$, resistance to ceftazidime–avibactam; ATM$^R$, resistance to aztreonam; C/T$^R$, resistance to ceftolozane–tazobactam; FEP$^R$, resistance to cefepime; IPM$^R$, resistance to imipenem; I-R$^R$, resistance to imipenem–relebactam; MEM$^R$, resistance to meropenem; MVB$^R$, resistance to meropenem–vaborbactam; TZP$^R$ resistance to piperacillin–tazobactam. Resistance to non-CZA β-lactam(s) was reported in brackets.
[c]According to MLST Pasteur scheme.
[d]$bla_{OXA-395}$, $bla_{OXA-486}$, $bla_{OXA-488}$, $bla_{OXA-494}$, $bla_{OXA-847}$, $bla_{OXA-848}$, $bla_{OXA-851}$ and $bla_{OXA-904}$ are members of the $bla_{OXA-50}$ family.
[e]PDC, *Pseudomonas aeruginosa*-derived cephalosporinase.
[f]CZA MIC values corresponding to a categorization of resistance are indicated in bold.

gene (a member of the $bla_{OXA-50}$ family) and the PDC-216 (Table 2). Similarly, all strains carrying VIM-1 and FIM-1 belonged to ST235 and carried the $bla_{OXA-488}$ gene (belonging to the $bla_{OXA-50}$ family) and the PDC-195. The FIM-1-producing strain (i.e., PSA8) also carried the $bla_{OXA-205}$ (a member of the $bla_{OXA-46}$ family) (Table 2).

The phylogenetic analysis revealed that the CZA$^R$ strains belonging to ST111 (55.5%, 10/18) showed a variable number of separating SNPs, exhibiting an SNP range 6–138 [mean, 61.4 (SD, 34.4); median, 53 (IQR, 40–69)], as well as those belonging to ST235 (33.3%, 6/18, including some ESBL-carrying strains reported below), which showed an SNP range 34–514 [mean, 307.9 (SD, 149.6); median, 290 (IQR, 211–467)], possibly supporting local transmission in a limited number of cases within ST111 (e.g., PSA62, PSA51, and PSA72) (Tables S2 and S3).

Overall, these results showed the presence of two major sequence types (STs) among CZA[R] strains, which were represented by two high-risk clones (i.e., ST111 and ST235), known to have a worldwide distribution and to be associated with epidemics and poor clinical outcomes (52). Moreover, the phylogenetic analysis of the two major STs clearly showed the presence of some clusters of highly similar strains carrying the same resistance mechanism in the same center, further underscoring the clonal nature of the population (i.e., the PSA72 and PSA62; the PSA1 and PSA7) (Fig. 1).

The remaining CZA[R] strains carried the $bla_{GES-1}$ (i.e., PSA9) and $bla_{PER-1}$ genes (i.e., PSA1, PSA7 and PSA38) and belonged to ST235, except for a PER-1-positive isolate (i.e., PSA38) belonging to ST643 (Table 2). GES-1 and PER-1 are both known to exhibit ESBL activity (5), and their involvement in the CZA resistance has been previously reported (40, 53–56). A recent study demonstrated that CZA resistance in *P. aeruginosa* can be attributed to the overexpression of $bla_{GES-1}$ mediated by the presence of a strong promoter (PcS) in class one integron located upstream of the $bla_{GES-1}$ gene (40). Sequence alignments revealed that the PSA9 strain described in our work (i.e., an ST235-type clinical isolate of *P. aeruginosa*) carried the $bla_{GES-1}$ gene in a class one integron containing a PcS [−35 (TTGACA) and −10 (TAAACT)] identical to that previously reported in *P. aeruginosa* strain chromosome sequences PA5083 (GenBank accession no. CP102174) (100% identity, 100% coverage), SE5352 (GenBank accession no. CP054843) (100% identity, 100% coverage), and 1903031130 (GenBank accession no. CP060392) (100% identity, 100% coverage) and associated to overexpression of $bla_{GES-1}$, which contributed to CZA resistance (40).

Concerning PER-1-producing strains, previous studies reported that the presence of $bla_{PER-1}$ gene may confer resistance to CZA in *P. aeruginosa*, demonstrating a poor activity of avibactam as an inhibitor toward PER-1 compared with other ESBLs (54–56). Noteworthy, these strains were also resistant to ceftolozane–tazobactam and cefiderocol (i.e., MIC >2 mg/L), which is in accordance with what was described both in *P. aeruginosa* (55) and in *Acinetobacter baumanii* (57, 58).

The majority of GES-1 and PER-1-producing strains included in the study also carried $bla_{OXA-488}$ (i.e., the PSA1, PSA7 and PSA9 strains) and $bla_{OXA-2}$ (i.e., the PSA1 and PSA7 strains) genes as well as the PDC-195 (i.e., the PSA1, PSA7 and PSA9 strains), whereas the strain PSA38 (i.e., a ST643 PER-1-producing *P. aeruginosa*) was the only one showing a different chromosomal β-lactamase content (i.e., $bla_{OXA-847}$ and PDC-212). Nevertheless, none of these chromosomal β-lactamases is associated with resistance to CZA, ceftolozane–tazobactam, and cefiderocol in *P. aeruginosa* (50, 59), further confirming the

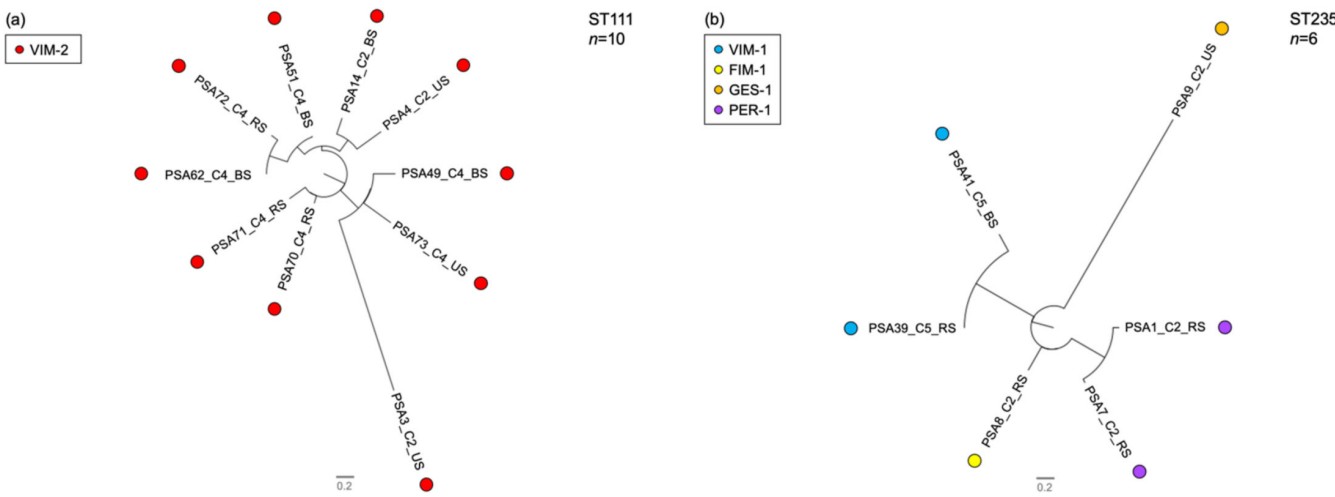

**FIG 1** Phylogenetic trees of the ceftazidime–avibactam resistant *P. aeruginosa* most-represented sequence types (STs): (a) ST111; (b) ST235. For each isolate, the sample code (PSA), the centrer code, (C) and the source (BS, blood sample; RS, respiratory sample; US, urinary sample) are reported. Filled circles of different colors identify the different types of transferable β-lactamases.

potential role of emerging acquired ESBLs (i.e., GES and PER types) in the *P. aeruginosa* CZA resistance.

Finally, only a CZA$^R$ isolate (i.e., the PSA10 strain) tested negative for carbapenemase genes and ESBL-coding genes, such as $bla_{PER-1}$ or $bla_{GES-1}$. However, this strain carried the $bla_{OXA-14}$ gene (i.e., an OXA-10 β-lactamase mutated) (Table 2), known to play a role in the *in vivo* acquisition of resistance to CZA and other cephalosporin–β-lactamase inhibitor combinations in *P. aeruginosa* (60), thus probably explaining the CZA resistance observed in this isolate. The PSA10 strain was also resistant to cefiderocol. Among class D β-lactamases, only OXA-427 has been recently demonstrated to have hydrolytic activity against cefiderocol (55), with cefiderocol-resistant OXA-427-producing *Enterobacterales* isolated in Belgium (61). Therefore, further investigations are needed to better elucidate the involvement of other OXA-like enzymes, such as OXA-14, in cefiderocol resistance in *P. aeruginosa*.

In addition, the OXA-14-producing *P. aeruginosa* isolate belonged to the emerging MDR *P. aeruginosa* clone ST664, which is considered one of the high-risk carbapenem-resistant clones in India and the cause of several outbreaks in China and Iran (62–64). The dissemination of this clone deserves of further attention.

Of note, no further carbapenemases or ESBLs other than VIM, FIM, GES, and PER types were detected. Moreover, no strain carrying more than one carbapenemase and/or ESBL was found.

Overall, these results provide a local update regarding acquired β-lactam resistance among MDR *P. aeruginosa* due to the emergence and spread of carbapenemases (65). Moreover, similarly to a previous work performed in Italy (66), the presence of two major clusters corresponding to international high-risk clones among CZA$^R$ strains adds further concern to our findings.

## β-lactamases and clonal characteristics of ceftazidime–avibactam-susceptible *P. aeruginosa* isolates

Of the thirty sequenced strains, eight were resistant to at least one β-lactam tested and susceptible to CZA. Genome analysis did not reveal the presence of carbapenemases and/or ESBLs. Several amino acid substitutions were found in OprD that likely contributed to the emergence of carbapenem resistance. Each strain harbored *oprD* mutations, including early terminations, resulting in complete porin loss (i.e., the PSA5, PSA23, and PSA64 strains), or substitutions (i.e., the PSA28, PSA37, PSA50, PSA60, and PSA63 strains).

Noteworthy, of the eight resistant strains, the MDR strain PSA5 showed resistance to aztreonam, cefepime, imipenem, meropenem, imipenem–relebactam, and piperacillin–tazobactam while retained susceptibility to CZA and ceftolozane–tazobactam (C/T) (Table 2; Table S1). This strain had two chromosomal β-lactamase genes, $bla_{PDC-195}$ and $bla_{OXA-488}$. The $bla_{OXA-488}$ gene is a member of the $bla_{OXA-50}$ family (59) that plays a minor role in β-lactam resistance (67). The PDC-195 protein carries several amino acid substitutions that likely do not result in an expanded-spectrum phenotype (68). However, the expression of $bla_{PDC-195}$ and $bla_{OXA-488}$ is linked to cell wall recycling and is induced by exposure to certain β-lactams (67, 69, 70). This strain (PSA5) belonged to ST253, an epidemic high-risk clone (71).

Interestingly, the strain PSA28 was resistant to amikacin, aztreonam, cefepime, piperacillin–tazobactam, tobramycin, and C/T but was susceptible to CZA (Table 2 and Table S1). This strain harbored the $bla_{OXA-486}$ gene, a variant of the intrinsic oxacillinase $bla_{OXA-50}$, which was associated with resistance to C/T in a study from Qatar (49).

The other members of the $bla_{OXA-50}$ family (i.e., $bla_{OXA-848}$, $bla_{OXA-851}$ and $bla_{OXA-904}$) were not considered as determinants of CZA resistance (72).

The eight strains resistant to at least one β-lactam tested and susceptible to CZA were all clonally unrelated, except for one minor cluster represented by two strains belonging to ST308 (i.e., *P. aeruginosa* PSA60 and PSA63) and exhibiting only 24 separating SNPs (data not shown).

Four multi-susceptible isolates (i.e., the PSA29, PSA44, PSA56, and PSA79 strains) were also sequenced for comparison. As shown in Table 2, these strains harbored no resistance mechanisms. Moreover, they were clonally unrelated each other and to the eight *P. aeruginosa* isolates resistant to at least one β-lactam tested and susceptible to CZA.

## Conclusions

In conclusion, our study demonstrated that ceftazidime–avibactam represents a highly active antipseudomonal β-lactam agent against MDR *P. aeruginosa*, allowing it to be considered a valid therapeutic option among the available antipseudomonal molecules, although the most active β-lactam remains the new siderophore cephalosporine cefiderocol. Among non-β-lactam agents, colistin and amikacin retain activity in a high proportion of isolates but caution against the use of these agents without the use of additional therapeutic measures should be considered (combination with another active agent).

However, our study presents some limitations. First, strains collected were obtained only from two Italian regions, in particular from the major centers of Apulia and a regional hospital in Basilicata. Moreover, not all CZA[R] isolates were sequenced (18/22). Nevertheless, the present data could be considered as a baseline for nationwide surveillance studies on the resistance mechanisms to ceftazidime–avibactam in *P. aeruginosa*.

We also demonstrated that resistance to ceftazidime–avibactam in *P. aeruginosa* correlates mostly with MBL production, and to a lesser extent with ESBL (GES and PER types) production. The presence of ceftazidime–avibactam-resistant high-risk clones ST111 and ST253 among different regional and extra-regional hospitals was also observed. Consequently, this scenario highlights the need for continuous surveillance programs to monitor local resistance and carbapenemase epidemiology of MDR *P. aeruginosa* as well as the prevalence of *P. aeruginosa* high-risk circulating clones.

## ACKNOWLEDGMENTS

This work was founded by a research grant from Pfizer Inc. (grant number ID#70161611). Article processing charges were partially paid with 5 x 1000 IRPEF funds in favour of the University of Foggia, in memory of Gianluca Montel. The authors recognize the financial contribution of European Union – NextGenerationUE as part of PNRR MUR – M4C2 – Investimento 1.3 - Public Call "Partenariati Estesi" - D.D. n. 341/2022.

## AUTHOR AFFILIATIONS

[1]Department of Clinical and Experimental Medicine, University of Foggia, Foggia, Italy
[2]Istituto Zooprofilattico Sperimentale della Puglia e della Basilicata, Foggia, Italy
[3]Clinical Pathology and Microbiology Unit, AOR San Carlo, Potenza, Italy
[4]Clinical Pathology and Microbiology Unit, Vito Fazzi Hospital, Lecce, Italy
[5]Clinical Pathology Unit, SS Annunziata Hospital, Taranto, Italy
[6]Department of Interdisciplinary Medicine, Microbiology Section, University of Bari Aldo Moro, Bari, Italy
[7]Microbiology and Virology Unit, AOU Policlinico Riuniti, Foggia, Italy
[8]IRCCS Fondazione Don Carlo Gnocchi ONLUS, Florence, Italy

## AUTHOR ORCIDs

Felice Valzano http://orcid.org/0000-0002-7638-1538
Fabio Arena http://orcid.org/0000-0002-7265-3698

## FUNDING

| Funder | Grant(s) | Author(s) |
|--------|----------|-----------|
| Pfizer Inc. | 70161611 | Fabio Arena |

## DATA AVAILABILITY

Draft genome sequences were deposited in GenBank under the BioProject PRJNA1026085.

## ADDITIONAL FILES

The following material is available online.

### Supplemental Material

**Supplemental material (Spectrum04266-23-S0001.pdf).** Tables S1 to S3.

### Open Peer Review

**PEER REVIEW HISTORY (review-history.pdf).** An accounting of the reviewer comments and feedback.

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
