## [Reviewer comments · Microbiology Spectrum]

Microbiology Spectrum

Resistance to ceftazidime-avibactam and other new β -lactams in *Pseudomonas aeruginosa* clinical isolates: a multi-center surveillance study

Felice Valzano, Gianfranco La Bella, Teresa Lopizzo, Anna Curci, Laura Lupo, Elisabetta Morelli, Adriana Mosca, Marianna Marangi, Raffaella Melfitano, Tiziana Rollo, Rosella De Nittis, and Fabio Arena

Corresponding Author(s): Fabio Arena, Universita degli Studi di Foggia

Review Timeline:

Submission Date:	January 10, 2024
Editorial Decision:	February 28, 2024
Revision Received:	April 5, 2024
Editorial Decision:	May 23, 2024
Revision Received:	May 23, 2024
Accepted:	June 9, 2024

Editor: Rafael Vignoli

Reviewer(s): Disclosure of reviewer identity is with reference to reviewer comments included in decision letter(s). The following individuals involved in review of your submission have agreed to reveal their identity: Marcela Alejandra Radice (Reviewer #2)

Transaction Report:

DOI: <https://doi.org/10.1128/spectrum.04266-23>

Re: Spectrum04266-23 (Resistance to ceftazidime-avibactam and other new β -lactams mediated by metallo- β -lactamases and extended-spectrum β -lactamases in *Pseudomonas aeruginosa* clinical isolates: a multi-center surveillance study)

Dear Prof. Fabio Arena:

Thank you for the privilege of reviewing your work. Your paper has been reviewed by two experts on the subject, and as you will see both suggest making changes to be publishable in Spectrum. Below you will find my comments, instructions from the Spectrum editorial office, and the reviewer comments.

Revision Guidelines

Sincerely,
Rafael Vignoli
Editor
Microbiology Spectrum

Reviewer #1 (Comments for the Author):

This study investigated the prevalence of resistance to GZA and comparators among *P. aeruginosa* clinical isolates in the context of a regional surveillance program. The topic is of interest, considering the potential use of GZA against MDR *P. aeruginosa*. Although most issues previously raised by Reviewers were sufficiently addressed, several shortcomings persist.

Major comments:

- the Introduction section is dramatically overlong (e.g. l.67-93), and the manuscript could certainly benefit from a shortened version of this section
- Characterization of GES-1-producing strain must be carefully revised. Authors claim a strong promoter (PcS) located upstream of the blaGES-1 gene could explain the observed CZA-R phenotype. However, sequence analysis was significantly flawed, since it cannot confirm the localization of blaGES-1 in a class 1 integron. Indeed, a BLAST analysis on PSA9 clearly showed int1 and blaGES-1 laying in different contigs. Still, the int1 sequence was in two different contigs. On top of this, please note that the sequence of the integron cassette promoter (Pc) in PSA9 was not that of a strong promoter, as currently reported in the text. Ref. 42 reported a PcS with a [-35]TTGACA....TAAACT[-10] sequence, but a [-35]TgGACA....TAAACT[-10] sequence was detectable instead, that is a Pc hybrid 1 promoter. Pc hybrid 1 was shown to drive a two-fold increase of blaGES transcription level, roughly (see doi:10.1128/AAC.00912-08). Please modify the text (methods, results) and referencing accordingly. I would also check for the presence of a P2 promoter.
- Authors should be aware that production of PER-1 has been previously associated with CZA resistance in *P. aeruginosa* CZA (https://doi.org/10.1128/aac.00339-23) (l.258-261). Furthermore, increased cefiderocol MIC among PER-producing *P. aeruginosa* have been reported previously (doi: 10.1128/aac.00828-22) (l.276-288). Results and conclusions must be carefully revised.
- (not mandatory) I would suggest to partially edit the title, as it appears too declamatory and anticipate somewhat expected results (MBL-mediated CZA resistance) that may inadvertently limit the readers' interest. E.g: "Resistance to ceftazidime-avibactam and other novel β -lactams in *Pseudomonas aeruginosa* clinical isolates: a multi-center surveillance study"

Specific comments:

- l.83. "generally found in the context of class 1 integrons", I would specify "in Pseudomonads"
- l.103 DTR phenotype?
- l.231: I would specify 18/22 CZA-R isolates
- error values for mean (SD) and median (IQR) should be provided (SNPs analysis)
- l.263: if overexpression, than should be blaGES-1 (vs overproduction related to the enzyme, GES-1)

Reviewer #2 (Comments for the Author):

In the present study, it was carried out the characterization of CZA resistant *P. aeruginosa* isolates recovered in 5 hospitals in Italy. Although not original, the results are relevant as they contribute to the knowledge in this critical pathogen, considering the few therapeutic options available for the treatment of infections caused by MDR strains. These data also contribute to the epidemiology of the region, in particular of Apulia.

I believe that this work would be of interest to the clinical and scientific community, although some changes are recommended for publication.

Major comments

The study is not representative of a surveillance study in the region since it only includes one hospital of the Basilicata region. Therefore, I suggest mentioning in the introduction that the aim of this study was to analyze the mechanisms which confer resistance to CZA, to characterized the resistant isolates (circulating clones), and to contribute to the epidemiology of the region. The selection of isolates subject to WGS is confusing. It is mentioned that CZA-resistant isolates were selected, but 18/22 of them were included. On the other hand, of those CZA-susceptible isolates, there were chosen 8 which presented resistance to at least one beta-lactam, and other sensitive ones. May be isolates within each group were randomly selected?

Lines 212 to 214: Analyzing the MIC values (MIC50 and MIC90) presented in Table 1, I consider that the differences between the MIC values for CZA respect those for CAZ or PTZ, are not enough to be highlighted in this paragraph. Even more so considering that the values are not absolute but, for example, >16. I suggest rewriting this paragraph.

Lines 247 to 251: Is there any SNP cutoff value to infer clonal relatedness. In my opinion, the SNPs values between some isolates are large enough to assume that these isolates are NOT "high clonal relatedness". Opposite to that mention in the text for both ST 111 and even more for ST235. See <https://www.ncbi.nlm.nih.gov/pmc/articles/PMC6851282/>

The "Abstract" as well as the "Importance" section should be written according to the suggested modifications.

Minor comments

Line 80: avoid repetition of "Furthermore"

Line 138: replace "Isolates from cystic fibrosis or pediatric patients" by "Isolates from patients with cystic fibrosis or pediatrics"

Line 163: replace "automatic" by "automated"

Line 171: replace "strain" by "bacterial genome"

Line 204: replace 51.6% by 51,7% and 15.9% by 15,8%

Line 287: add: cefiderocol and cefotolozane /tazobactam.

Line 289: it must be added: tested negative for carbapenemase "and ESBL coding genes such as blaPER-1 or blaGES-1"

Line 305: according to previous comments, It would be better to refer to the fact that the study contributes to local epidemiology

Line 609: replace "originating from" by "recovered from"

Table 2: Replace "Transferable resistance genes" by "Transferable Beta-lactamase coding genes"

Supplementary data: Delete table 1 as antibiotic abbreviations are included in the text and it is not clear what do you refer with the column "MIC ranges".

Dear Editor,

we thank you and the Reviewers for the careful revision that helped us in improving our manuscript. Please find below a point-by-point response to the comments raised by the Reviewers. A marked-up version of the manuscript with revisions highlighted in response to the issues raised by the Reviewers has also been uploaded for your convenience.

Response to Reviewers

Reviewer #1 (Comments for the Author):

This study investigated the prevalence of resistance to CZA and comparators among *P. aeruginosa* clinical isolates in the context of a regional surveillance program. The topic is of interest, considering the potential use of CZA against MDR *P. aeruginosa*. Although most issues previously raised by Reviewers were sufficiently addressed, several shortcomings persist.

Major comments:

1. The Introduction section is dramatically overlong (e.g. l.67-93), and the manuscript could certainly benefit from a shortened version of this section.

Authors' response: We are grateful for your suggestion. We have reduced the length of the Introduction section accordingly (lines 66–78).

2. Characterization of GES-1-producing strain must be carefully revised. Authors claim a strong promoter (PcS) located upstream of the bla_{GES-1} gene could explain the observed CZA-R phenotype. However, sequence analysis was significantly flawed, since it cannot confirm the localization of bla_{GES-1} in a class 1 integron. Indeed, a BLAST analysis on PSA9 clearly showed intI1 and bla_{GES-1} laying in different contigs. Still, the intI1 sequence was in two different contigs. On top of this, please note that the sequence of the integron cassette promoter (Pc) in PSA9 was not that of a strong promoter, as currently reported in the text. Ref. 42 reported a PcS with a [-35]TTGACA...TAAACT[-10] sequence, but a [-35]TgGACA...TAAACT[-10] sequence was detectable instead, that is a Pc hybrid 1 promoter. Pc hybrid 1 was shown to drive a two-fold increase of bla_{GES} transcription level, roughly (see doi:10.1128/AAC.00912-08). Please modify the text (methods, results) and referencing accordingly. I would also check for the presence of a P2 promoter.

Authors' response: We are grateful to Reviewer 1 for raising this point and for the careful revision. As correctly underscored by Reviewer 1, in the PSA9 draft genome (Illumina sequencing technology), intI1 and bla_{GES-1} were in different contigs. Therefore, we decided to perform Nanopore sequencing and we carried out a hybrid assembly of the *P. aeruginosa* PSA9 genome (an update of the new *de novo* assembly has been deposited at NCBI and is now available with the accession number CP150132). The genome is now closed (the chromosome and the plasmid) and we can now certainly state that bla_{GES-1} gene is integrated in the chromosome and is located immediately downstream of an intI1 integron carrying a strong promoter (i.e., PcS), which has been previously associated with the overexpression of bla_{GES-1} (Li *et al.*, 2023 – doi: 10.1016/j.drug.2023.100973), thus explaining the high CZA MIC observed with this strain (see lines 252–260). We have modified the methods section adding information about Nanopore sequencing and hybrid assembly performed for the PSA9 strain (see lines 175–186).

3. Authors should be aware that production of PER-1 has been previously associated with CZA resistance in *P. aeruginosa* CZA (https://doi.org/10.1128/aac.00339-23) (l.258-261). Furthermore, increased ceftiderocol MIC among PER-producing *P. aeruginosa* have been reported previously (doi: 10.1128/aac.00828-22) (l.276-288). Results and conclusions must be carefully revised.

Authors' response: We are grateful to reviewer 1 for providing us with these additional references that have been now included in the paper in support of the PER-1 involvement in ceftiderocol resistance (see lines 250–251, 261–263, 266–273).

4. (not mandatory) I would suggest to partially edit the title, as it appears too declamatory and anticipate

somewhat expected results (MBL-mediated CZA resistance) that may inadvertently limit the readers' interest. E.g: "Resistance to ceftazidime-avibactam and other novel β -lactams in *Pseudomonas aeruginosa* clinical isolates: a multi-center surveillance study".

Authors' response: We agree with the Reviewer's suggestion. We modified the title accordingly.

Specific comments

1. 1.83. "generally found in the context of class 1 integrons", I would specify "in Pseudomonads"
2. - 1.103 DTR phenotype?
3. 1.231: I would specify 18/22 CZA-R isolates.
4. error values for mean (SD) and median (IQR) should be provided (SNPs analysis).
5. 1.263: if overexpression, than should be blaGES-1 (vs overproduction related to the enzyme, GES-1).

Authors' response: We made all these suggested changes (see lines 219, 220, 237–240, 253).

Reviewer #2 (Comments for the Author):

In the present study, it was carried out the characterization of CZA resistant *P. aeruginosa* isolates recovered in 5 hospitals in Italy. Although not original, the results are relevant as they contribute to the knowledge in this critical pathogen, considering the few therapeutic options available for the treatment of infections caused by MDR strains. These data also contribute to the epidemiology of the region, in particular of Apulia. I believe that this work would be of interest to the clinical and scientific community, although some changes are recommended for publication.

Mayor comments

The study is not representative of a surveillance study in the region since it only includes one hospital of the Basilicata region. Therefore, I suggest mentioning in the introduction that the aim of this study was to analyze the mechanisms which confer resistance to CZA, to characterized the resistant isolates (circulating clones), and to contribute to the epidemiology of the region.

Authors' response: Thanks for your comment. We have modified the text accordingly (see lines 111–114).

The selection of isolates subject to WGS is confusing. It is mentioned that CZA-resistant isolates were selected, but 18/22 of them were included. On the other hand, of those CZA-susceptible isolates, there were chosen 8 which presented resistance to at least one beta-lactam, and other sensitive ones. May be isolates within each group were randomly selected?

Authors' response: We are grateful to reviewer 2 for the careful revision. Unfortunately, 4/22 CZA-resistant strains were not sequenced since they were not culturable after -80°C storage. Thus, only 18 strains were included in the WGS collection. Regarding CZA-susceptible strains, we decided to select strains susceptible to CZA and resistant to ceftolozane/tazobactam and/or at least one carbapenem (including the carbapenem- β -lactamase inhibitor combinations); unfortunately, also in this category, 7 strains weren't viable after storage and thawing. Finally, only the strains with a non-resistant pattern were selected randomly. We have modified the text to better explain the selection criteria of isolates subjected to WGS (see lines 144–149).

Lines 212 to 214: Analyzing the MIC values (MIC50 and MIC90) presented in Table 1, I consider that the differences between the MIC values for CZA respect those for CAZ or PTZ, are not enough to be highlighted in this paragraph. Even more so considering that the values are not absolute but, for example, >16. I suggest rewriting this paragraph.

Authors' response: We agree with the Reviewer 2 and decided to delete this paragraph.

Lines 247 to 251: Is there any SNP cutoff value to infer clonal relatedness. In my opinion, the SNPs values between some isolates are large enough to assume that these isolates are NOT "high clonal relatedness". Opposite to that mention in the text for both ST 111 and even more for ST235. See <https://www.ncbi.nlm.nih.gov/pmc/articles/PMC6851282/>

Authors' response: Thank you for raising this point. To set a SNP cutoff value is important to investigate the bacterial transmission within and between clinical settings of the same hospital in an outbreak investigation, which is not the scope of this study. We agree that a "high clonal relatedness" could be considered an overstatement in absence of clear cut-off values. However, we are sufficiently confident that the range of 6 to 138 (mean 61.4 [SD 34.4]; median 53 [IQR 40-69]) separating SNPs for ST111, as well as the range of 34 to 514 (mean 307.9 [SD 149.6]; median 290 [IQR 211-467]) for ST235, cannot be overlooked and can represent a clonal relatedness within the same sequence type. Considering that, we slightly modified the text, also according to "specific comment" #4 of Reviewer 1 (see lines 237–240).

The "Abstract" as well as the "Importance" section should be written according to the suggested modifications.

Minor comments

Line 80: avoid repetition of "Furthermore"

Line 138: replace "Isolates from cystic fibrosis or pediatric patients" by "Isolates from patients with cystic fibrosis or pediatrics"

Line 163: replace "automatic" by "automated"

Line 171: replace "strain" by "bacterial genome"

Line 204: replace 51.6% by 51,7% and 15.9% by 15,8%

Line 287: add: cefiderocol and cefotolozane /tazobactam.

Line 289: it must be added: tested negative for carbapenemase "and ESBL coding genes such as blaPER-1 or blaGES-1"

Line 305: according to previous comments, It would be better to refer to the fact that the study contributes to local epidemiology

Line 609: replace "originating from" by "recovered from"

Table 2: Replace "Transferable resistance genes" by "Transferable Beta-lactamase coding genes"

Supplementary data: Delete table 1 as antibiotic abrevations are included in the text and it is not clear what do you refer with the column "MIC ranges".

Authors' response: Suggestions have been accepted and modifications done (see lines 122, 150, 158, 195, 264, 275, 291, 594 and Table 2).

Re: Spectrum04266-23R1 (Resistance to ceftazidime-avibactam and other new β -lactams in *Pseudomonas aeruginosa* clinical isolates: a multi-center surveillance study)

Dear Prof. Fabio Arena:

Thank you for the privilege of reviewing your work. On this occasion, both reviewers have agreed that your manuscript has been considerably improved but that small modifications are still necessary for its final acceptance. Below you will find my comments, instructions from the Spectrum editorial office, and the reviewer comments.

Revision Guidelines

Sincerely,
Rafael Vignoli
Editor
Microbiology Spectrum

Reviewer #1 (Comments for the Author):

The current version of the manuscript has clearly benefited from suggested revisions, and previously raised issues have been sufficiently addressed.

I would just call for a more carefully evaluation of separating SNPs within ST235 and ST111 isolates, since the span of reported ranges (mean, median) does not really support their are "closely related".

It could be worth stating the SNPs separating each isolate pair within the same ST are quite variable, possibly supporting local transmission in a limited number of cases within ST111 (e.g. PSA62_C4_BS, PSA51_C4_BS, PSA72_C4_BS)

Reviewer #2 (Comments for the Author):

Lines 146 to 149: Correct the selection criteria. Selection criteria 1 is not mentioned.

line 68: Replace "the constitutive AmpC-type" for "the chromosomally encoded Ampc-type" or the "intrinsic Ampc-Type"

Dear Editor,

we thank you and the Reviewers for the reviewing process of the article: “Resistance to ceftazidime-avibactam and other new β -lactams in *Pseudomonas aeruginosa* clinical isolates: a multi-center surveillance study”.

Please find below the point-by-point response (in blue) to the comments raised by Reviewer 1 and 2. A marked-up version of the manuscript with changes highlighted according to Reviewers comments has also been uploaded for your convenience.

Response to Reviewers

Reviewer #1 (Comments for the Author):

The current version of the manuscript has clearly benefited from suggested revisions, and previously raised issues have been sufficiently addressed.

I would just call for a more carefully evaluation of separating SNPs within ST235 and ST111 isolates, since the span of reported ranges (mean, median) does not really support their are "closely related".

It could be worth stating the SNPs separating each isolate pair within the same ST are quite variable, possibly supporting local transmission in a limited number of cases within ST111 (e.g. PSA62_C4_BS, PSA51_C4_BS, PSA72_C4_BS)

Authors' response: We agree with the Reviewer 1. We modified the text accordingly (lines 236–241).

Reviewer #2 (Comments for the Author):

Lines 146 to 149: Correct the selection criteria. Selection criteria 1 is not mentioned.

Authors' response: We have double checked. The selection criteria 1 is stated at line 144.

line 68: Replace "the constitutive AmpC-type" for "the chromosomally encoded Ampc-type" or the "intrinsic Ampc-Type".

Authors' response: We modified the text as suggested by Reviewer 2.

Re: Spectrum04266-23R2 (Resistance to ceftazidime-avibactam and other new β -lactams in *Pseudomonas aeruginosa* clinical isolates: a multi-center surveillance study)

Dear Prof. Fabio Arena:

I am pleased to inform you that your manuscript has been accepted. Now I am forwarding it to the ASM production staff for publication. Your paper will first be checked to make sure all elements meet the technical requirements. ASM staff will contact you if anything needs to be revised before copyediting and production can begin. Otherwise, you will be notified when your proofs are ready to be viewed.

Sincerely,
Rafael Vignoli
Editor
Microbiology Spectrum